# Effect of Non-Linear Properties of Intermediate Layer on Sliding Contact of Homogeneous and Coated Elastic Solids

Elena V. Torskaya * and Fedor I. Stepanov

Ishlinsky Institute for Problems in Mechanics RAS, 119526 Moscow, Russia; stepanov_ipm@mail.ru
* Correspondence: torskaya@mail.ru; Tel.: +7-495-434-2090

**Abstract:** Lubrication in the contact area can be considered as a viscoelastic layer, especially in the presence of particles in it, or under conditions of low temperatures. The properties of this layer are often non-linear, in particular, they depend on local pressure. The paper presents a formulation and numerical-analytical method for solving the contact problem in the presence of viscoelastic layers, the compliance of which depends on the applied pressure and is included in the formulation as the corresponding operator. The layer is homogeneous or coated elastic half-space. For the selected type of operator, the influence of parameters, which characterize the nonlinearity of the model, on the distribution of contact pressure and the coefficient of friction due to hysteresis losses was analyzed. It is shown that for the nonlinear model, the maximum contact pressures are higher, and the friction coefficient is lower than for the linear model with constant compliance. The effect of non-linearity for a wide range of sliding velocities is considered. An analysis of principal shear and tensile-compressive stresses for a homogeneous elastic half-space and for a coating, in particular, for a coating-substrate interface, was also carried out.

**Keywords:** contact problem; coating; viscoelastic layer; stresses; nonlinear Maxwell model



## 1. Introduction

A contact problem with thin viscoelastic layers on the surface arises when the contact occurs in the presence of a 'third body'. This could be a fretting process [1] or lubricated contact [2,3] under conditions of high pressure in the contact area [4–8], as well as relatively low temperatures [9], in which oil lubricants begin to behave like solid viscoelastic bodies.

In cases where the thickness and/or modulus of elasticity of the layer is relatively large, 3D models of material are used [10–13], in which the layer is considered as an isotropic viscoelastic body. When the layer is relatively soft and thin, it is possible to use simplified one-dimensional viscoelastic models, including the Maxwell model [2,3,14,15], the Kelvin–Voigt model [16–19], and the standard viscoelastic body model [20,21]. More information about the viscoelastic contact can be found in review [22].

In references [14,15], the rolling of cylinders was considered in the presence of a lubricant. The effect of thin surface layers of lubricant on contact characteristics and internal stresses was analyzed. To describe the properties of surface layers in the normal and tangential directions, the one-dimensional model of Maxwell viscoelastic layer was used [14,15], and the lubricant flow between the surfaces was described using the Reynolds equation [15].

In reference [2], the mutual sliding of two discs was modeled in the presence of high-pressured lubricant. To simulate the viscoelastic properties of an oil film in a sliding contact, a one-dimensional nonlinear (in tangential direction) Maxwell model was proposed. This model is in good agreement with the Eyring fluid flow theory. In reference [3], a similar model was used, differing only in the form of the nonlinear function.

The rolling of an elastic sphere over an elastic half-space covered with a viscoelastic layer was studied in references [17,18]. The viscoelastic layer was modeled using the

one-dimensional Kelvin model in tangential direction and a modified Winkler model for the normal contact. The distribution of stick and slip zones in the contact, as well as stresses in the contacting bodies, was calculated and analyzed [18].

In ref [19], the possibility of modeling an elastohydrodynamic contact using a nonlinear one-dimensional Kelvin−Voigt model was considered. The model was developed based on an elastic one-dimensional model obtained in references [23–25] from the Navier-Cauchy equation, assuming that the lubricant thickness is small. The model allows for consideration of the local absence of lubrication; therefore, several contact conditions were considered including mixed lubrication.

The rolling contact of a rigid sphere over an elastic half-space covered with a viscoelastic layer (simulating a lubricant layer) was studied in reference [21]. Two approaches were proposed: a one-dimensional model of a standard viscoelastic body and a complete model based on the Navier-Cauchy equations using Papkovich−Neiber potentials. It was shown that for effective modeling of elastohydrodynamic contact, it is necessary to use pressure-dependent viscoelastic models.

In this paper, we study the sliding contact of homogeneous and inhomogeneous elastic bodies in the presence of a viscoelastic layer in the contact area. A one-dimensional nonlinear Maxwell model is proposed, in which the layer compliance depends on the local pressure. The aim of the study was to develop an adequate method for solving such problems, as well as to analyze the influence of the nonlinearity factor on contact and internal stresses, and on the friction force arising due to sliding resistance.

## 2. Problem Formulation

Let us consider the motion of a smooth slider of arbitrary shape over the boundary of homogeneous or covered elastic half-space, on the surface of which there is a viscoelastic layer of thickness $h_3$ (Figure 1). A normal force, $Q$, acts on the slider. The layer simulates an intermediate medium, which may consist of lubricant wear (or abrasive) particles. In references [14,15], the mechanical properties of the layer are described using the viscoelastic Maxwell model:

$$\dot{w}_3(x,y) = C\left(\frac{p(x,y)}{T} + \dot{p}(x,y)\right), \tag{1}$$

where $w_3$ are the normal displacements of the layer boundary, $C$ is the constant compliance of the layer, $T$ is the relaxation time, and $p(x,y)$ is the contact pressure.

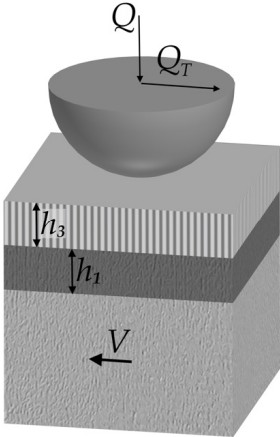

**Figure 1.** Scheme of sliding contact.

This model has unlimited creep, which in some cases describes well the properties of the intermediate medium. At the same time, it is logical to assume that the compliance of the layer should depend on the pressure, for example, due to the fact that the layer becomes

thinner and, therefore, stiffer in the area of high pressures. In this regard, the following modification of (1) is proposed:

$$\dot{w}_3(x,y) = C[p(x,y)]\left(\frac{p(x,y)}{T} + \dot{p}(x,y)\right),$$ (2)

Here, the dependence of compliance on pressure is determined by the specific features of the intermediate layer.

The sliding contact occurs at a constant velocity, $V$, so the problem is quasi-static. To provide sliding at the constant velocity over the viscoelastic layer, an a priori unknown constant tangential force, $Q_T$, is required, equal to the sliding resistance force due to the rheological properties of the layer. Let us position the Cartesian coordinate system, associated with the slider, in such a way that the $XY$ plane coincides with the undeformed upper surface of the viscoelastic layer and the $Z$ axis is normal to the layer surface. In this case, the contact condition will look like this:

$$\begin{array}{c} w_1(x,y) + w_3(x,y) = f(x,y) - D \quad (x,y) \in \Omega \\ \sigma_z = 0 \qquad (x,y) \notin \Omega \\ \tau_{xz} = 0, \ \tau_{yx} = 0 \end{array}$$ (3)

Here, $w_1(x,y)$ are the vertical displacements of the upper surface of a homogeneous or covered elastic half-space, $f(x,y)$ is a function that describes the shape of the slider, $D$ is the penetration of the slider into the half-space, and $\Omega$ is the contact area. The material properties of a homogeneous half-space are determined by Young's modulus $E_2$ and Poisson's ratio $\nu_2$. In the case of coated half-space, we have $E_i$ and $\nu_i$, where $i = 1, 2$ corresponds to the coating and the half-space, respectively. In the "spring" model under consideration, the normal stresses at the upper boundary of the layer ($z = 0$) and at the boundary between the layer and the elastic half-space ($z = h_3$) have the same values, since the stresses do not change along the length of the spring.

It is assumed that the surface layer has the function of lubrication, so the friction force is not taken into account in the formulation of the contact problem (3).

An equilibrium condition is also satisfied:

$$Q = \iint\limits_{\Omega} p(x,y)dxdy,$$ (4)

Since we are considering a smooth slider, the zero-pressure condition must be satisfied at the boundary of the contact area $\Omega$.

In the presence of a coating of thickness $h_1$, it is also necessary to formulate the conditions at the interface between the coating and the half-space:

$$\begin{array}{c} w^{(1)} = w^{(2)}, \ u_x^{(1)} = u_x^{(2)}, \ u_y^{(1)} = u_y^{(2)}, \\ \sigma_z^{(1)} = \sigma_z^{(2)}, \ \tau_{xz}^{(1)} = \tau_{xz}^{(2)}, \ \tau_{yz}^{(1)} = \tau_{yz}^{(2)}. \end{array}$$ (5)

Here, $\sigma_x^{(i)}$, $\tau_{xz}^{(i)}$, $\tau_{yz}^{(i)}$ are normal and tangential stresses; $w^{(i)}$, $u_x^{(i)}$, $u_y^{(i)}$ are normal and tangential displacements. Relations (5) correspond to the conditions of full adhesion at the coating-substrate interface, when the continuity of normal and tangential stresses and displacements takes place.

It is required that the contact area, the distribution of contact pressure and the slider penetration are found.

## 3. Method of Solution

The solution was obtained using the boundary element method and an iterative procedure. A rectangular area $\Omega^* \supset \Omega$ is selected, on which a mesh of $n$ identical square elements of size $2a \times 2a$ is constructed. The shape of the slider is represented as a set of

values $f_i$ $(i = 1 \ldots n)$ which are $f(x, y)$ for the center of each square element; the resulting pressure is obtained as a piecewise function $p_i$ $(i = 1 \ldots n)$. Knowing the dependences of the vertical displacements of the layer boundary on the uniformly distributed pressure inside a square element of the surface, we can transform Equations (3) and (4) into a system of $n + 1$ linear algebraic equations:

$$
\begin{bmatrix}
\begin{pmatrix}
A_{11} & \cdots & A_{1n} \\
\vdots & \ddots & \vdots \\
A_{n1} & \cdots & A_{nn}
\end{pmatrix} & \begin{matrix} 1 \\ \vdots \\ 1 \end{matrix} \\
\begin{pmatrix} 4a^2 & \cdots & 4a^2 \end{pmatrix} & 0
\end{bmatrix}
\cdot
\begin{bmatrix} p_1 \\ \vdots \\ p_n \\ D \end{bmatrix}
=
\begin{bmatrix} f_1 \\ \vdots \\ f_n \\ Q \end{bmatrix},
\tag{6}
$$

The last equation here is the equilibrium condition (4) in terms of boundary elements: $A$ is a matrix of influence coefficients, which are boundary displacements related to constant pressure inside each square element. They are obtained by superposition of the displacements of a viscoelastic layer and a homogeneous (covered) half-space:

$$
A_{ij} = A_{ij}^{(3)} + A_{ij}^{(1)},
\tag{7}
$$

where $A_{ij}^{(3)}$ corresponds to the normal displacements of the viscoelastic layer, and $A_{ij}^{(1)}$ corresponds to a homogeneous or covered half-space. Using direct integration of (2), and taking into account that non-zero displacements are only on the part of the layer surface that was or is under the influence of distributed pressure, we obtain the following expressions for $A_{ij}^{(3)}$:

$$
\begin{aligned}
A_{ij}^{(3)} &= 0, \ (x_j - x_i) > a, -a > (y_j - y_i) > a \\
A_{ij}^{(3)} &= -C_i \left( 1 + \frac{a - (x_j - x_i)}{TV} \right), -a < (x_j - x_i) < a, -a < (y_j - y_i) < a \\
A_{ij}^{(3)} &= -C_i \left( \frac{2a}{TV} \right), \ (x_j - x_i) < -a, -a < (y_j - y_i) < a
\end{aligned}
\tag{8}
$$

Here, $C_i$ are determined by the form of the operator $C$ for a point with $(x, y)$ coordinates.

The matrix coefficients $A_{ij}^{(1)}$ are calculated using the relations for displacements of the elastic half-space boundary [26]:

$$
A_{ij}^{(1)} = -\frac{(1 - v_1^2)}{\pi E_1} \iint\limits_{\Omega_i} \frac{1}{\sqrt{(x_j - \xi)^2 + (y_j - \eta)^2}} d\xi d\eta,
\tag{9}
$$

For a two-layer elastic half-space, the matrix coefficients $A_{ij}^{(1)}$ are obtained using a method based on double integral Fourier transforms [27]:

$$
\begin{aligned}
A_{ij}^{(1)} = -\frac{a(1 + v_1)}{\pi E_1} \int\limits_0^{\pi/2} \int\limits_0^\infty \Delta(\gamma, \varphi, \lambda, E_1, E_2) \cos\big((x_i - x_j)\gamma \cos(\varphi)/a\big) \\
\cos\big((y_i - y_j)\gamma \sin(\varphi)/a\big) d\gamma d\varphi
\end{aligned}
\tag{10}
$$

here $\gamma$, $\varphi$ are coordinates in the space of double Fourier transforms; $\lambda = h_1/a$, $\Delta(\gamma, \varphi, \lambda, E_1, E_2)$ is an analytic function, which is obtained from boundary conditions (formulated for uniformly loaded surface elements) using the Fourier transforms.

Using an iterative procedure, at each step of which system (4) is solved for $A_{ij} = A_{ij}^{(1)}$ and the contact conditions (including zero pressure at the boundary of the contact zone) are satisfied, the unknown contact area, pressure distribution, and penetration $D$ are obtained

for an elastic body without the viscoelastic layer. Next, an iterative procedure is carried out in order to obtain a solution based on the following type of $C[p(x,y)]$:

$$C[p(x,y)] = C_0\left(1 + \delta - \frac{2\delta}{1 + e^{-p/p_0}}\right), \tag{11}$$

This type of operator is chosen from the following considerations. In the absence of pressure ($p = 0$), the compliance of the layer is $C_0$. The action of pressure leads to a decrease in the thickness of the layer in the contact zone; it is assumed that the layer thickness at infinite pressure tends towards a finite value (including zero). Thus, the minimum compliance of the layer is defined as $C_0(1 - \delta)$, where $\delta$ is a dimensionless parameter. The parameter $p_0$ has the dimension in Pascal and determines the sensitivity of the layer properties to changes in the contact pressure. Figure 2 illustrates the model for different values of $p_0$ and $\delta$.

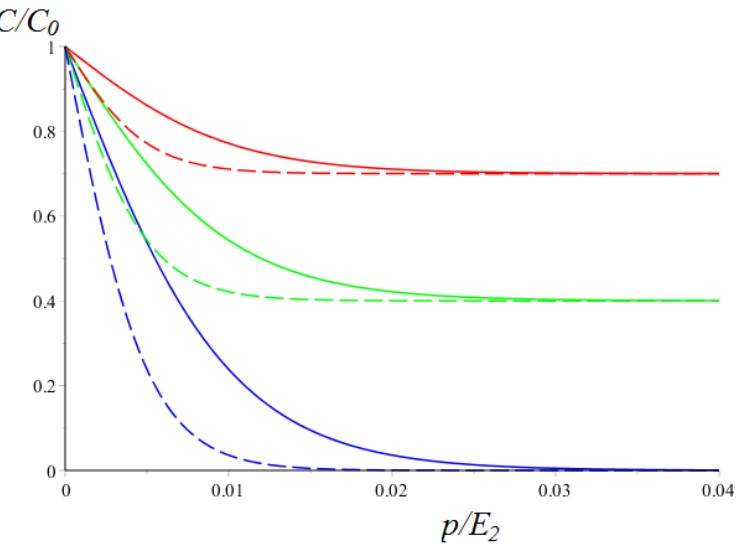

**Figure 2.** Compliance of the layer depending on contact pressure: $\delta = 0.3$ (red curve), $\delta = 0.6$ (green curve), $\delta = 1.0$ (blue curve); $p'_0 = 0.0025$ (dashed), $p'_0 = 0.005$ (solid).

The calculated contact pressure was used to find internal stresses in the coated elastic half-space. The method for calculating stresses in a two-layered elastic half-space is based on double integral Fourier transforms [10]. The expressions for such calculations were obtained for layered elastic half-space in reference [28] and used, for example, in reference [10].

The method was used to calculate tensile-compressive ($\sigma_x$) stresses in the coated half-space.

The method for calculating stresses in a two-layered elastic half-space is based on double integral Fourier transforms. In the case of homogeneous half-space, the Love solution [29] was used for the constant pressure distributed inside a surface element.

The resulting stresses (both for the coated and the homogeneous half-space) are obtained by superposition.

## 4. Results and Discussion

The following dimensionless parameters are used to analyze the results for a spherical slider with radius $R$:

$$(x',y',z') = (x,y,z)/R, \ C' = C_0 \cdot E_2/R, \ Q' = Q/(R^2 E_2),$$
$$p'(x',y') = p(x',y')/E_2, \ V' = VT/R, \ E'_1 = E_1/E_2, \ p_0' = p_0/E_2,$$
$$\sigma'_{x,y,z} = \sigma_{x,y,z}/E_2, \ \tau'_1 = \tau_1/E_2.$$

Let us present the results obtained for an elastic half-space with a layer for which the properties are described by relation (11). Here, and further, the sliding direction is positive (along the $0x'$ axis). The sliding velocity was chosen for reasons of visual demonstration of the influence of viscoelastic properties on the characteristics of the contact. Figure 3 shows the distribution of the contact pressure and the compliance of the layer in the contact area. The contact pressure distribution is asymmetric, which is typical for the sliding contact of materials with rheological properties. The contact spot has a shape close to a semicircle. The compliance of the layer near the center of the contact area is close to minimal value.

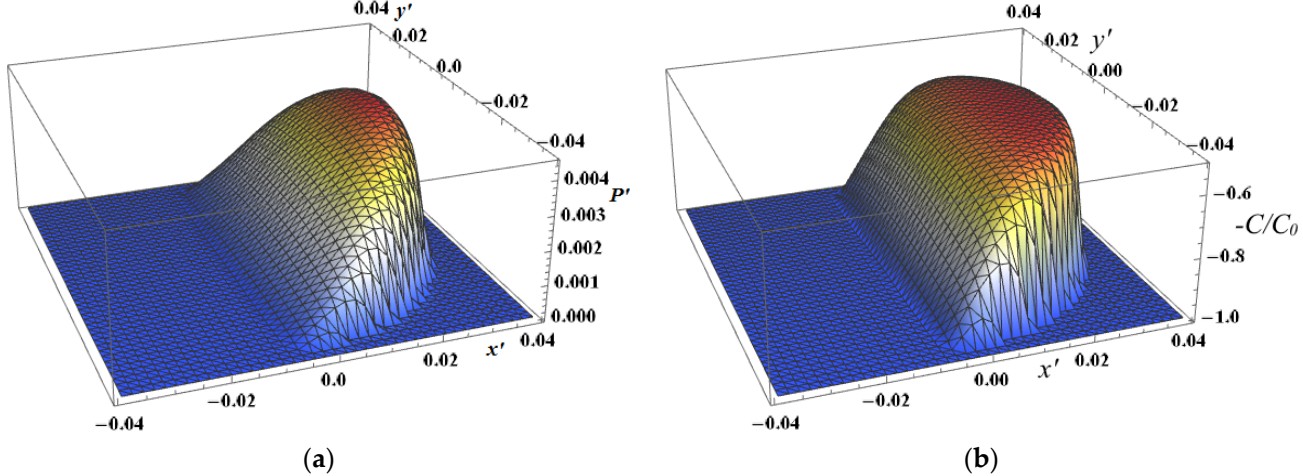

(a)                                                              (b)

**Figure 3.** Distribution of the contact pressure and the corresponding distribution of the layer compliance (substrate−half-space). Calculation parameters: $Q' = 5 \times 10^{-6}$; $V' = 10^{-3}$; $\nu_2 = 0.3$; $C' = 0.02$; $\delta = 0.5$, $p'_0 = 0.001$. (a)—distribution of contact pressure. (b)—distribution of compliance.

The parameters $p_0$, $\delta$ describing the effect of local pressure on the properties of the layer in (11) also significantly affect the contact pressure distribution. The influence of $p_0$ is illustrated by the curves in Figure 4. Solid curves of contact pressure distribution lie between two dashed curves relating to linear models with the maximum and minimum compliance allowed by the non-linear model (11). The smaller the value of $p_0$, the greater the maximum contact pressure and the smaller the size of the contact area becomes.

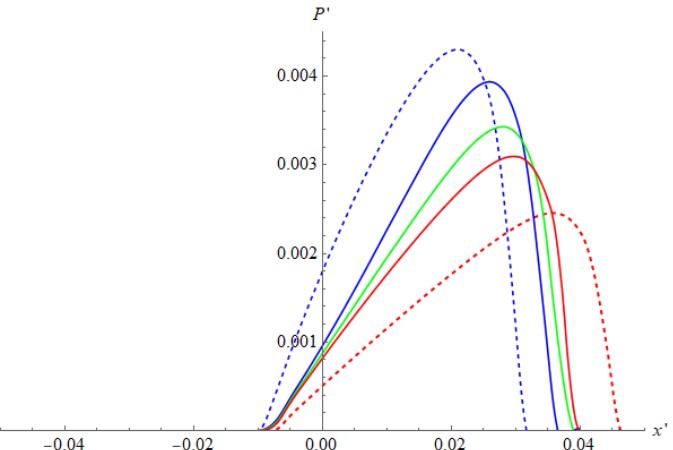

**Figure 4.** Contact pressure distribution along the X-axis. Calculation parameters: $Q' = 5 \times 10^{-6}$; $V' = 10^{-3}$; $\nu_2 = 0.3$; $C' = 0.02$; $\delta = 0.5$, $p'_0 = 0.0025$ (blue curve), $p'_0 = 0.005$ (green curve), $p'_0 = 0.05$ (red curve). Linear model: $C' = 0.02$ (red dashed curve), $C' = 0.01$ (blue dashed curve).

A similar analysis was made for the parameter $\delta$ (Figure 5). The fact that the contact pressure increases with the increase in $\delta$ is predictable. Interestingly, the blue curve,

calculated for the complete absence of the layer in the limiting case of infinite pressure, is more symmetric, i.e., the influence of the rheological properties of the layer decreases with increasing $\delta$.

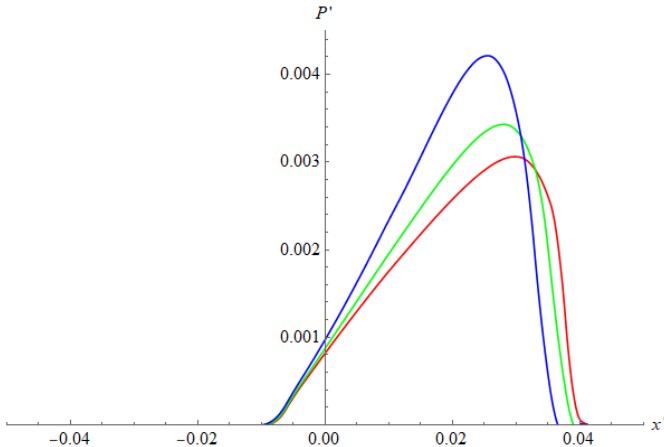

**Figure 5.** Distribution of contact pressure depending on parameter $\delta$. Calculation parameters: $Q' = 5 \times 10^{-6}$; $V' = 10^{-3}$; $\nu_2 = 0.3$; $C' = 0.02$ (layer compliance); $p'_0 = 0.005$; $\delta = 0.01$ (red curve), $\delta = 0.5$ (green curve), $\delta = 1$ (blue curve).

The sliding resistance is directly related to the asymmetry of the contact pressure. The moment of forces and the coefficient associated of which are defined as:

$$M = \iint_{\Omega} x \cdot p(x,y)dxdy, \quad \mu = \frac{\iint_{\Omega} x \cdot p(x,y)dxdy}{RQ}, \quad Q_T = -\mu Q, \tag{12}$$

Since the layer has lubricating properties, the sliding resistance due to viscosity can make a significant contribution to the total frictional force. Below is an analysis of the influence of the input parameters of the model on the value of $\mu$.

Figure 6 shows curves illustrating the influence of the relative compliance of the layer on the value. In this case, a decrease in the relative compliance of the layer is provided by increasing the rigidity of the half-space at a fixed load. The greater the rigidity of the half-space, the more significant the effect of the layer on the contact conditions. This explains the increase in the value of $\mu$, in which the asymptote probably tends to the value typical for the case of contact between the slider and the layer on a rigid base. It should be noted that the curves obtained for the linear and non-linear models are similar in shape and slope at each point, especially for relatively large values of $C'$. The same can be said about the curves illustrating the effect of the thickness of relatively rigid coatings on $\mu$ (Figure 7), since an increase in thickness leads to an increase in the integral rigidity of a two-layered elastic body.

It is interesting to analyze the effect of the non-linearity factor on the friction coefficient as a function of sliding velocity (Figure 8). For a 3D model of the viscoelastic layer [10], as well as for the standard viscoelastic body model [20], this dependence usually is non-monotonic. For the Maxwell model the friction coefficient decreases with the velocity increase. Generally, for the non-linear model the friction coefficient is smaller than for linear one. The effect depends on the model parameters and the sliding velocity. The difference between $\mu(V)$ dependences for the linear and non-linear models of the layer (Figure 8b) demonstrates maximum at a fixed value of velocity. The value depends on the model parameters. The coefficient of friction here is significantly higher than for the results from the previous figures, since in this case a larger load value, $Q'$, was used for calculations.

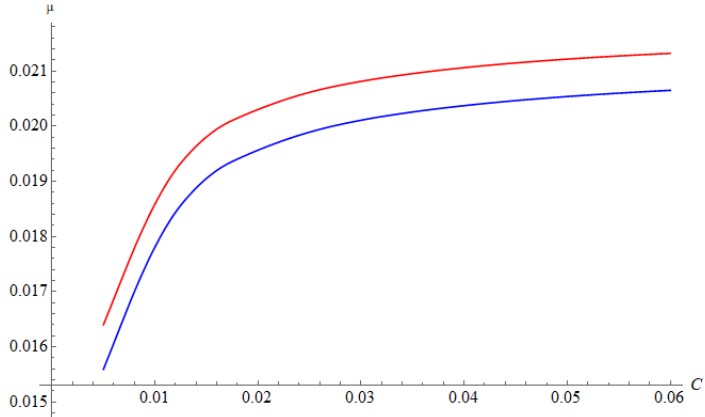

**Figure 6.** The dependence of the coefficient $\mu$ on the relative compliance of the layer. Calculation parameters: $Q' = 5 \times 10^{-6}$; $V' = 10^{-3}$; $\nu_2 = 0.3$; $C' = 0.02$. Linear compliance—red curve; $p'_0 = 0.005$, $\delta = 0.5$ —blue curve.

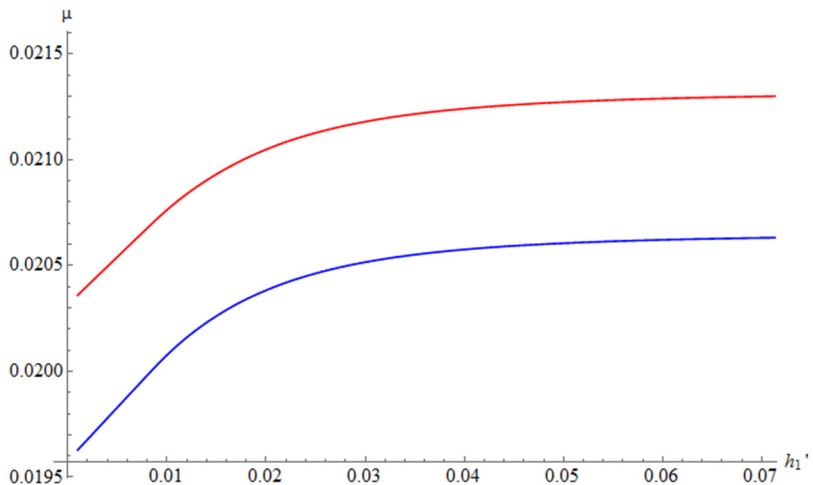

**Figure 7.** Dependence of the friction coefficient on the thickness of rigid coating ($E_1/E_3 = 3$). Calculation parameters: $Q' = 5 \cdot 10^{-6}$; $V' = 10^{-3}$; $\nu_1 = 0.3$; $\nu_2 = 0.3$; $\delta = 0.5$; linear compliance—red curve; $p'_0 = 0.005$ —blue curve.

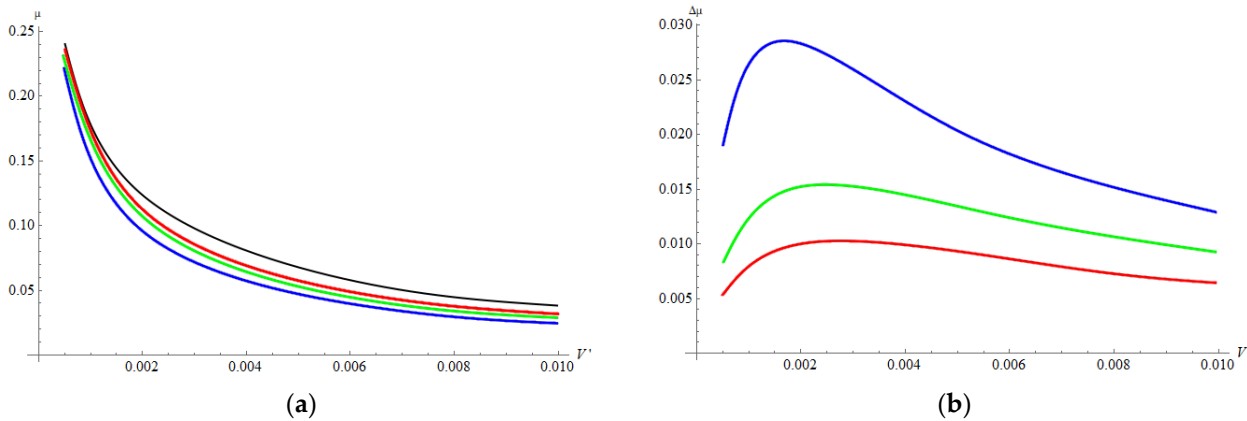

**(a)**                                                                **(b)**

**Figure 8.** (**a**) Dependence of the friction coefficient on sliding velocity (viscoelastic layer on the homogeneous elastic half-space). (**b**) Difference of the friction coefficient between linear and non-linear models depending on sliding velocity. Calculation parameters: $Q' = 25 \times 10^{-6}$; $\nu_2 = 0.3$; $C' = 0.02$; $\delta = 0.5$; $p'_0 = 0.0025$ (blue curves), $p'_0 = 0.005$ (green curves), $p'_0 = 0.0075$ (red curves); linear model—black curve.

Further analysis concerns the stresses that arise in an elastic half-space during sliding. The tensile and principal shear stresses used in most strength criteria were considered. Figure 9 shows the distributions of tensile-compressive stresses on the surface of an elastic half-space in the presence of a viscoelastic layer characterized by different values of $\delta$. For a homogeneous half-space, this component of the stress tensor has maximum value on the surface. The compressive stresses in this case are negative, and the tensile stresses are positive. The tension maximum takes place at the boundary of the contact spot. As $\delta$ increases, the maximum values of both tensile and compressive stresses increase. It should be noted that, in contrast to the case of a viscoelastic half-space [30], the presence of a layer does not lead to an instantaneous change in the stress sign at the edge of the contact area, although the gradient of the curves near the frontal part of the contact area is quite large.

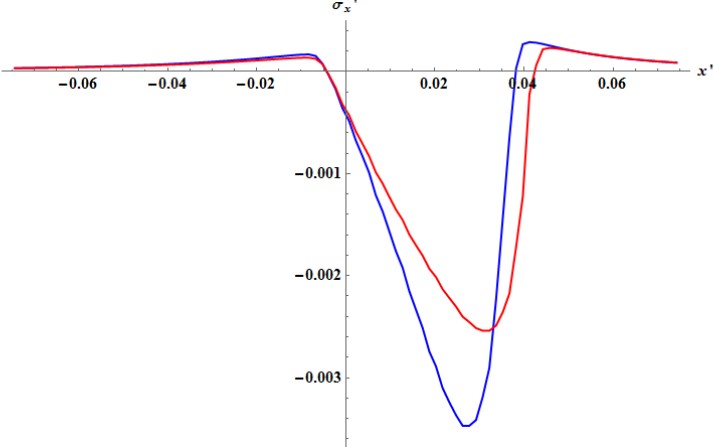

**Figure 9.** Distribution of the $\sigma_x$ stress along the X-axis. Calculation parameters: $Q' = 5 \times 10^{-6}$; $V' = 10^{-3}$; $\nu_2 = 0.3$; $C' = 0.02$; $p'_0 = 0.005$; $\delta = 0.01$ (red curve), $\delta = 1$ (blue curve).

The principal shear stresses are calculated for a plane parallel to the sliding direction and passing through the point of initial contact (Figure 10). The shape of isolines is typical for frictional contact. An increase in the value of $\delta$, which characterizes the degree of nonlinearity of the layer properties, from 0.1 to 1.0 leads to an increase in the maximum stress values of almost 35 percent.

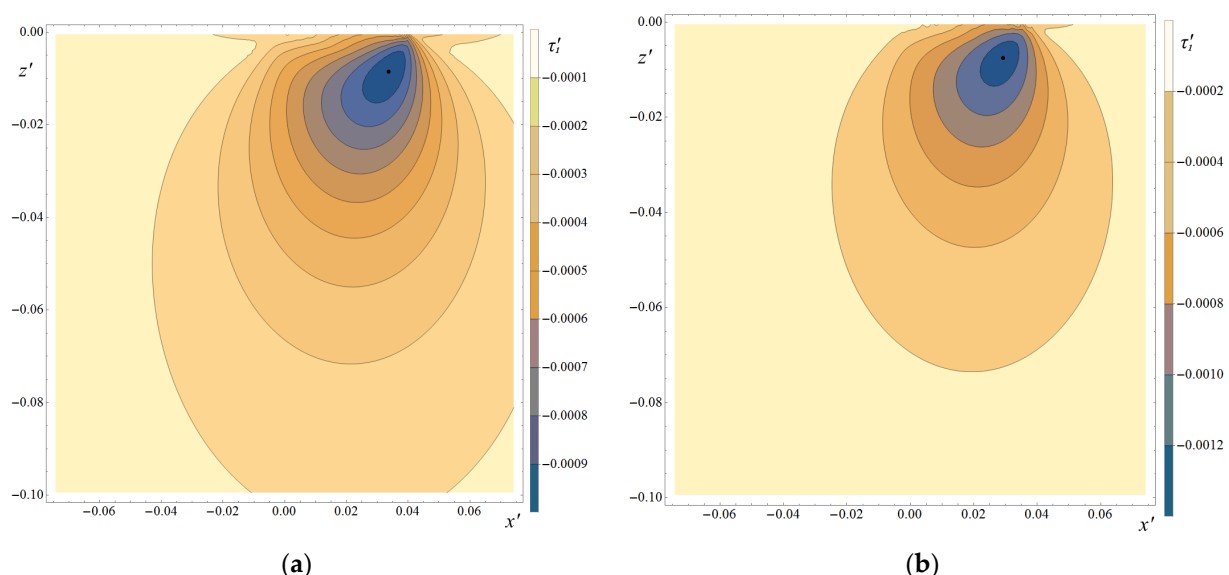

(**a**)                                                                 (**b**)

**Figure 10.** Distribution of maximum shear stresses in the XZ plane. Calculation parameters: $Q' = 5 \times 10^{-6}$; $V' = 10^{-3}$; $\nu_2 = 0.3$; $C' = 0.02$; $p'_0 = 0.005$; (**a**) $\delta = 0.01$, (**b**) $\delta = 1$; (**a**) $\tau_1^{max}/E_2 = -0.000982$; (**b**) $\tau_1^{max}/E_2 = -0.00132$.

In the presence of coatings, it is important to evaluate the stresses not only at the surface but also at the interface, which often has poor strength properties. Distributions of tensile-compressive stresses were obtained for a plane parallel to the sliding direction and passing through the point of initial contact (Figure 11) for the coatings with different thickness (using parameters identical to those in the non-linear model). The non-linearity of the layer properties is manifested in an increase in the maximum contact pressure: for a thinner coating by 12, and for a thicker one by 16 percent. The left picture illustrates the case of relatively thin coating, in which the thickness is almost 40 times less than characteristic size of contact zone in $0x'$ direction. The stresses inside the coating are almost the same as at the surface. For the case of the thicker coating (Figure 11b), the coating bend with compression at the surface and tension at the coating-substrate interface occurs. Figure 12 shows the distributions of tensile-compressive stresses for coatings of different thicknesses on the surface and at the interface. Since there is a jump of stresses at the interface, two curves are obtained for it (for coating and for substrate materials). The dotted lines are the distributions obtained for the linear model of the viscoelastic layer, and the solid lines are for the nonlinear model. The non-linearity factor significantly affects the distribution of surface stresses for the thicker coating but has almost no effect on the stresses at the interface. In the case of thin coating, the influence of this factor is approximately the same for the surface and the interface in percentage terms. For the relatively thick coating, maximal tension occurs at the coating-substrate interface under the contact spot. It means that brittle cracks can be initiated at the interface. For the case of relatively thin coatings, the maximal tension is at the surface. It should also be noted that for thin coatings there is a sharp change in tension-compression, which can be dangerous in terms of the formation of vertical cracks coming from the surface.

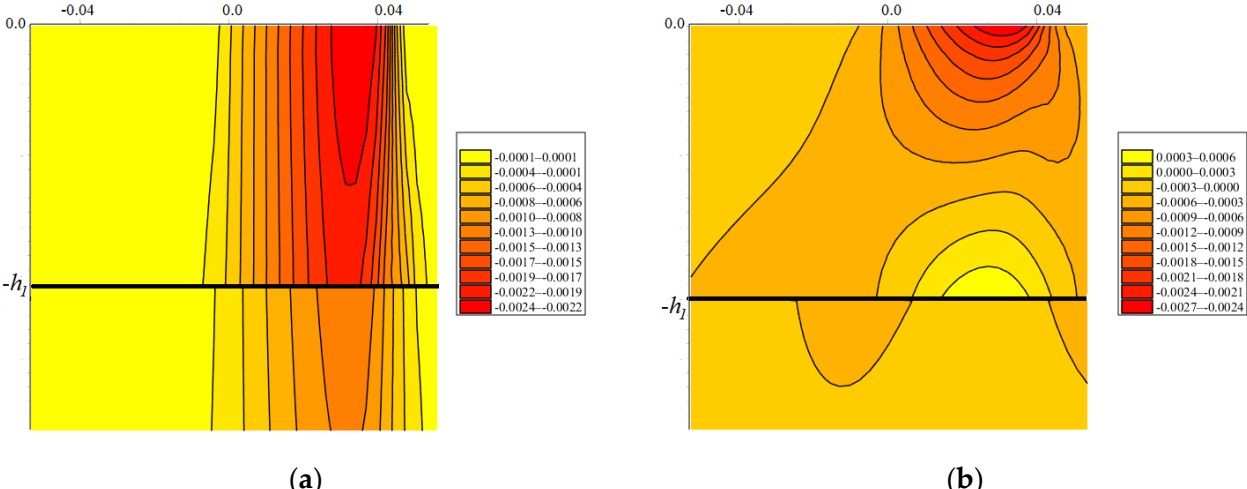

(**a**)   (**b**)

**Figure 11.** Distribution of tensile-compressive stresses in the half space and the intermediate layer. Calculation parameters: (**a**) $Q' = 5 \times 10^{-6}$; $V' = 10^{-3}$; $\nu_1 = 0.3$; $\nu_2 = 0.3$; $C' = 0.02$; $h'_1 = 0.001$, $h'_1 = 0.02$; (**b**) linear compliance.

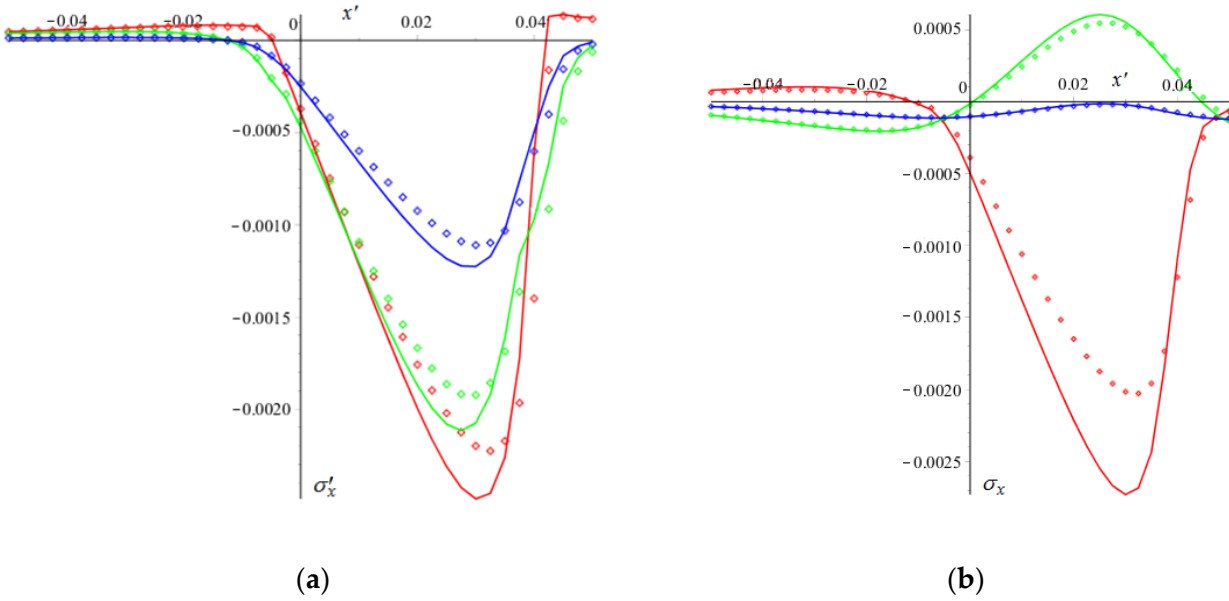

       **(a)**                                                     **(b)**

**Figure 12.** Distribution of tensile-compressive stresses. Calculation parameters: $Q' = 5 \times 10^{-6}$; $V' = 10^{-3}$; $\nu_1 = 0.3$; $\nu_2 = 0.3$; $C' = 0.02$; (**a**) $h'_1 = 0.001$, (**b**) $h'_1 = 0.02$; linear compliance—point curves; $\delta = 0.5$; $p'_0 = 0.005$ —solid curves; the coating surface—red curves; coating-substrate interface: coating side—green curves; substrate side—blue curves.

## 5. Conclusions

The developed model makes it possible to estimate the influence of the nonlinearity factor, which manifests itself in the dependence of the compliance of the viscoelastic Maxwell layer on pressure on the quasi-static sliding of a smooth indenter over the viscoelastic layer lying on a homogeneous or two-layered elastic body. A numerical-analytical method for solving the problem has been developed, which can be used for various types of operators relating compliance to local contact pressure. The results obtained for the selected operator, corresponding to the physical decrease in the layer thickness under the action of pressure, were analyzed. It is shown that in this case the nonlinearity factor leads to an increase in the maximum values of the contact pressure and some decrease in the size of the contact area. The study of the influence of the input parameters of the model on the coefficient of friction due to the resistance to sliding by the viscoelastic material showed that for the non-linear model this coefficient is smaller, with all other parameters being equal. The difference between the coefficient for linear and non-linear models depends on sliding velocity non-monotonically. The more rigid (relative to the viscoelastic layer) a homogeneous or two-layer half-space is, the greater the value of the coefficient. An analysis of the stresses in the elastic half-space under the layer showed an increase in their maximum values when the nonlinearity of the viscoelastic layer was taken into account. For bodies with rigid coatings, it is shown that the nonlinearity factor significantly affects surface stresses, and for relatively thin coatings, it also affects stresses at the interface.

**Author Contributions:** Conceptualization, E.V.T.; methodology, E.V.T. and F.I.S.; formal analysis, E.V.T.; investigation F.I.S.; data curation, E.V.T.; writing—original draft preparation, E.V.T.; writing—review and editing, E.V.T. and F.I.S.; visualization, F.I.S.; supervision, E.V.T.; project administration, E.V.T. All authors have read and agreed to the published version of the manuscript.

**Funding:** This research was supported by the Government program (Project Reg. No. 123021700050-1) and partially supported by RFBR, grant number 21-58-52006.

**Data Availability Statement:** Not applicable.

**Conflicts of Interest:** The authors declare no conflict of interest.

## Nomenclature

| | |
|---|---|
| $x, y, z$ | coordinates in the Cartesian coordinate system; |
| $Q$ | normal load applied to the slider; |
| $Q_T$ | tangential force applied to the slider; |
| $V$ | sliding velocity; |
| $D$ | penetration of the slider |
| $f(x,y)$ | shape of the slider; |
| $R$ | radius of the slider; |
| $P$ | contact pressure; |
| $\Omega$ | contact area; |
| $A$ | half size of the mesh element; |
| $w_i, i = 1; 3$ | vertical displacement of the upper layer $(i = 3)$ and the intermediate layer $(i = 1)$; |
| $h_i, i = 1; 3$ | layer thickness : $(i = 1)$—coating, $(i = 3)$—viscoelastic layer; |
| $E_i, i = 1; 2$ | Young modulus of the half-space $(i = 2)$ and the intermediate layer $(i = 1)$; |
| $\nu_i, i = 1; 2$ | Poisson ratio of the half-space $(i = 2)$ and the intermediate layer $(i = 1)$; |
| $C$ | compliance of the viscoelastic layer; |
| $T$ | relaxation time; |
| $p_0, \delta$ | parameters of nonlinear compliance; |
| $\sigma_{x,y,z}, \tau_{xy,xz,yz}$ | normal and tangential stresses; |
| $\tau_1$ | principal shear stress; |
| $M$ | moment acting on the slider; |
| $\mu$ | coefficient of sliding resistance (friction); |
| $w^{(i)}, u_x^{(i)}, u_y^{(i)}$ $(i = 1, 2)$ | normal and tangential displacements at the intermediate layer $(i = 1)$—half space $(i = 2)$ interface. |

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
