# Peer review of "Effect of Non-Linear Properties of Intermediate Layer on Sliding Contact of Homogeneous and Coated Elastic Solids"

_lubricants, doi:10.3390/lubricants11080333_

Round 1

Reviewer 1 Report

Please, see attached file.

English very difficult to understand and incomprehensible. Many phrases have no meaning, e.g., "an experimental effect of the high pressure was the linear dependence", smooth slider of arbitrary shape at a constant velocity V", "the nonlinearity of the model on the distribution of contact pressure", and many others.

Author Response

As for English, the phrases mentioned by the reviewer have been edited.

Reviewer 2 Report

The manuscript in general is well-written. However, some of the presentations are quite confusing and thus unfriendly to a reader. I recommend the acceptance after the following problems are addressed.

1. I am confused with the first paragraph of Introduction. The authors tried to enumerate several cases and they actually listed two cases:  third-body and lubricated cases. Here there is a logic loophole: Lubricant is also a third-body. I guess, that the idea, which the authors intended to convey in this first paragraph, the oil/lubricant in some scenario can be solid-like rather than a fluid.    

2. About Eq. (4). When you say the equilibrium conditions, you should define what Q is!. It is the total external compression force.

3. In Eq. (5), the expression of z=h_3+h_1 is not a good idea because you later use z to indicating an axis direction. And say more on the physical meanings of Eq. (5), such as the continuity of stress and displacements can be of quite help to a reader.

4. Eq. (6) is also perplexing: A is the influence/compliance matrix connecting p_i (pressure ) to displacement/profile f)i. While D is penetration displacement and Q is compression. And you see that p_i-f_i and D-Q relations are reversed! What is the point to combine p_i +D together? Furthermore, the physical meaning of and how to compute s_i in Eq. (6) are not given.

5. In Results and discussion, there is a statement on line 168: The contact pressure distribution is asymmetric, which is typical for sliding contact of materials with rheological properties.” I am confused with this statement. In the equilibrium equation of Eq. (4), there is only normal/vertical external force of Q and there is no horizontal/tangential force, and it is a statics (and thus no dynamic horizontal friction force). My question is: Why the contact pressure distribution is symmetric? Again, in Eq. (4) or (6), there is no horizontal force causing sliding.

Author Response

Thank you very much for your kind and very professional review. We agree with all comments and have made the necessary changes to the text:

  1. I am confused with the first paragraph of Introduction. The authors tried to enumerate several cases and they actually listed two cases: “ third-body” and “lubricated” cases. Here there is a logic loophole: Lubricant is also a “third-body”. I guess, that the idea, which the authors intended to convey in this first paragraph, the oil/lubricant in some scenario can be solid-like rather than a fluid.

We agree with the remark. The first paragraph of the Introduction has been changed.

  1. About Eq. (4). When you say the equilibrium conditions, you should define what Q is!. It is the total external compression force.

We've added an explanation.

  1. In Eq. (5), the expression of z=h_3+h_1 is not a good idea because you later use z to indicating an axis direction. And say more on the physical meanings of Eq. (5), such as the continuity of stress and displacements can be of quite help to a reader.

We agree, since the coordinate changes with deformation.

  1. Eq. (6) is also perplexing: A is the influence/compliance matrix connecting p_i (pressure ) to displacement/profile f)i. While D is penetration displacement and Q is compression. And you see that p_i-f_i and D-Q relations are reversed! What is the point to combine p_i +D together? Furthermore, the physical meaning of and how to compute s_i in Eq. (6) are not given.

Here our mistake is that there is no explanation for s_i. These are the areas of the elements (replaced in the formula, since in our case they are constant). Thus, the last line in the matrix gives the equation of equilibrium, without which the system for determining pressure and penetration D would be incomplete.

  1. In Results and discussion, there is a statement on line 168: “The contact pressure distribution is asymmetric, which is typical for sliding contact of materials with rheological properties.” I am confused with this statement. In the equilibrium equation of Eq. (4), there is only normal/vertical external force of Q and there is no horizontal/tangential force, and it is a statics (and thus no dynamic horizontal friction force). My question is: Why the contact pressure distribution is symmetric? Again, in Eq. (4) or (6), there is no horizontal force causing sliding.

Thanks for this comment. The equilibrium condition is formulated only for the normal force. For viscoelastic bodies, there is always a resistance to sliding, so a tangential force must be applied to ensure sliding at a constant speed. We have added Figure 1 where this force is. It is unknown in advance, depends on the velocity, and is determined by solving the contact problem in terms of the moment due to the asymmetry of the contact pressure (see (12)).

Reviewer 3 Report

Comments on the manuscript: lubricants-2508965-peer-review-v1

Effect of non-linear properties of intermediate layer on sliding contact of homogeneous and coated elastic solids”.

Dear authors,

The manuscript entitled “Effect of non-linear properties of intermediate layer on sliding contact of homogeneous and coated elastic solids”  is a very interesting theoretical work in the field of mechanical contacts. The proposed model of compliance described by Eq. (11) is very interesting. Also,  the variation of the friction coefficients presented in Fig. 7 are according to the real behaviours.

 We consider that following first two observations can improve the quality of the manuscript and the last observation must be operated in the manuscript, being a mistake.

1. We consider that, for more clarity, will be necessary a general figure with a smooth slider over a half-space of which is a viscoelastic layer having an imposed thickness h3, and with the coordinate system (X,Y,Z) and the direction of the sliding speed V. Also will be better to be indicated the deformation wi.

2. In Eq. (1) and (2) we  suspect that the models are based on derivative of displacement and pressure, and C has unit m^3/N.

3. In Nomenclature M is indicated as tangential force but in  Eq. (14) M is a moment. Must be corrected!

Finally, we appreciate this manuscript and consider that can be published in Lubricants.  

Author Response

The authors are very grateful for the evaluation of the study and helpful comments.

Below are our answers.

  1. We consider that, for more clarity, will be necessary a general figure with a smooth slider over a half-space of which is a viscoelastic layer having an imposed thickness h3, and with the coordinate system (X,Y,Z) and the direction of the sliding speed V. Also will be better to be indicated the deformation wi.

Thanks for this idea. New Figure 1 added.

  1. In Eq. (1) and (2) we suspect that the models are based on derivative of displacement and pressure, and C has unit m^3/N.

You are right. In (1) and (2) we denoted the time derivatives with dots.

  1. In Nomenclature M is indicated as tangential force but in Eq. (14) M is a moment. Must be corrected!

Thank you. We corrected it.

Round 2

Reviewer 1 Report

Authors took into account almost all remarks of the review.